# Generation of a Realistic Synthetic Laryngeal Cancer Cohort for AI Applications

**DOI:** 10.3390/cancers16030639

**Published:** 2024-02-01

**Authors:** Mika Katalinic, Martin Schenk, Stefan Franke, Alexander Katalinic, Thomas Neumuth, Andreas Dietz, Matthaeus Stoehr, Jan Gaebel

**Affiliations:** 1Innovation Center Computer Assisted Surgery, Faculty of Medicine, University Leipzig, 04109 Leipzig, Germany; mika.katalinic@medizin.uni-leizig.de (M.K.);; 2Institute of Social Medicine and Epidemiology, University of Luebeck, 23562 Luebeck, Germany; alexander.katalinic@uksh.de; 3Department of Otolaryngology, Head and Neck Surgery, University Hospital Leipzig, 04103 Leipzig, Germany

**Keywords:** synthetic data, machine learning, laryngeal cancer, electronic patient record, oncological decision support

## Abstract

**Simple Summary:**

The use of synthetic patient data can help address patient privacy concerns and the general availability of clinical data. It can overcome the challenges associated with obtaining real patient data for use in medical research, healthcare analytics, and clinical decision support. We propose an approach to provide synthetic patient data for laryngeal cancer, a relatively rare but complex disease. We adapted an existing synthesis technology to produce realistic prevalence and age distributions in the generated patient datasets. We verified the methodology and validated the results using real patient data from a German cancer registry.

**Abstract:**

Background: Obtaining large amounts of real patient data involves great efforts and expenses, and processing this data is fraught with data protection concerns. Consequently, data sharing might not always be possible, particularly when large, open science datasets are needed, as for AI development. For such purposes, the generation of realistic synthetic data may be the solution. Our project aimed to generate realistic cancer data with the use case of laryngeal cancer. Methods: We used the open-source software Synthea and programmed an additional module for development, treatment and follow-up for laryngeal cancer by using external, real-world (RW) evidence from guidelines and cancer registries from Germany. To generate an incidence-based cohort view, we randomly drew laryngeal cancer cases from the simulated population and deceased persons, stratified by the real-world age and sex distributions at diagnosis. Results: A module with age- and stage-specific treatment and prognosis for laryngeal cancer was successfully implemented. The synthesized population reflects RW prevalence well, extracting a cohort of 50,000 laryngeal cancer patients. Descriptive data on stage-specific and 5-year overall survival were in accordance with published data. Conclusions: We developed a large cohort of realistic synthetic laryngeal cancer cases with Synthea. Such data can be shared and published open source without data protection issues.

## 1. Introduction

### 1.1. Motivation for Synthetic Data Generation

The use of synthetic patient data has become increasingly important in recent years to overcome the challenges associated with obtaining real patient data for use in medical research, healthcare analytics, and clinical decision support. The ability to generate realistic and diverse patient data can help to advance medical research and improve healthcare outcomes. Historically, the benefits of analyzing collected patient data have been significant, including increased accuracy, timely diagnosis, and the discovery of new knowledge about the disease and its progression. However, there are several challenges associated with obtaining real patient data for use in medical research and healthcare analytics. These challenges include [1]:Privacy concerns: Real patient data often contains sensitive personal information that must be protected to ensure patient privacy;Limited availability: Real patient data is often difficult to obtain in large quantities, particularly for specific populations or conditions;High costs: Obtaining real patient data can be costly, particularly when it requires the use of special equipment or specialized personnel;Compliance of patients to approve of the utilization of their personal data.

Synthetic patient data can help us to overcome these challenges by providing a way to generate realistic and diverse patient data that can be used for a variety of applications. Synthetic data can be generated in large quantities and can be customized to meet specific research or clinical needs. In addition, synthetic data do not have the same privacy concerns as real patient data, because they are not associated with any specific individuals [2]. Furthermore, the creation of electronic health records can support and facilitate the design and implementation of clinical trials. By not using real patient data, processing limitations can be avoided. It is possible to modulate the course of studies at different points by changing variables. This results in a low-cost, saving time. and flexible way of planning studies and conducting them in a realistic setting [3]. Synthetically generated patient data has also proven useful in the development of artificial intelligence (AI) applications and learning algorithms. In this case, it is often necessary to know the exact distributions of the variables in the training dataset used and to have the ability to change the distributions at will in order to test and verify the functionality of the AI. Those prerequisites are not fully given in real datasets, whereas in synthetic datasets every used distribution is known and can easily be altered if necessary. Ive et al. applied synthetic data to successfully train a model for natural language processing for the generation of discharge reports [4]. Yiang et al. and Das et al. applied synthetic CT images to help classify COVID-19 patients in a cohort of respiratory diseases [5,6]. Levine et al. used synthetic data to train a neural network for diagnosing ovarian cancer [7].

### 1.2. Synthetic Patient Population Simulator

Synthea is an open-source synthetic patient data generation tool, developed to generate large datasets of synthetic patient data [8]. The tool uses a combination of algorithms and real-world data to simulate patient histories, including demographics, medical conditions, and treatments. Each synthesized clinical record contains a patient’s entire medical history in the form of structured items (e.g., clinical encounter (any interaction between patient and healthcare provider), observations). Each item can reference any clinical dictionary or nomenclature to express the medical semantics. The generated data can be customized to meet specific research or clinical needs, and can also be used to test and validate healthcare systems and applications [9]. One of the key advantages of Synthea is that it can generate data that are representative of real-world populations, which can help to ensure that the data are generalizable to real-world scenarios. Additionally, the software allows for the generation of data with specific characteristics, such as age, race, and medical conditions, as carried out by Chen et al. for lung cancer [2]. This makes it a useful tool for researchers and healthcare professionals who need to study specific populations or test the performance of healthcare systems in specific scenarios. For instance, Bala et al. generated medical notes using Synthea, which were translated to plain natural language by AI [10]. They demonstrated the benefits of patient empowerment and showed how the patient–clinical relationship can be improved. Participants felt that they could better manage their own health and technology, which was well-received. Walonoski et al. used Synthea to synthesize COVID-19 data. Since it was applied quite early during the coronavirus pandemic, the datasets were mostly used for online challenges and hackathons [9]. In the early months of the coronavirus pandemic, those data helped healthcare professionals to exchange ideas about clinical developments, prediction, and testing. They were used in hackathons and conferences. However, it was too early to be applied in clinical delivery. Scalfani et al. analyzed a regulatory aspect of healthcare. They created synthetic patient records to analyze how healthcare policy impacts breast cancer survival rates [11].

There are other open-source and commercial synthetic data generators available [12]. However, many approaches to synthetic data generation focused on detailed clinical settings, e.g., the investigation of pathophysiology, gene expression, or neuronal structure, and are therefore too specific. The majority are based on machine learning approaches, e.g., GAN-based, which requires an appropriate amount of training data to be available [13]. Others are too general in scope to produce realistic EHR [9]. Some projects with a general approach are the Synthetic Electronic Medical Records Generator (EMERGE) [14] and the medical Generative Adversarial Network (medGAN) [15]. We chose Synthea because it allowed us to manually integrate clinical pathways and expert knowledge. This ensures that the clinical items which are necessary for analysis are available. The advantage of Synthea encompasses the potential of data synthesis without an initial data set. Only real statistics on disease or treatment frequencies are required [9]. The output format helps us to further process and apply the generated data. The community aspect of allowing different expert groups to contribute clinical models allows for a suitable dissemination of our results.

### 1.3. Clinical Use Case of Laryngeal Cancer

Laryngeal cancer (ICD-10: C32) has an annual incidence of approximately 184,000 cases worldwide, but is a relatively rare disease (age-standardized incidence rate of 3.6 and 0.5 per 100,000 in men and women, respectively) [16]. In Germany, approximately 3400 people are diagnosed with laryngeal cancer each year [17,18]. The five-year survival rate is 56% and 59% in men and women, respectively [18]. The main risk factors for laryngeal cancer are chronic tobacco and alcohol abuse, especially in combination [19]. Other risk factors also include occupational causes such as exposure to asbestos, ionizing radiation, or exposure to coal products and tar products [20]. Clinical treatment guidelines in head and neck oncology, such as the National Comprehensive Cancer Network on Head and Neck cancers or the German S3 Guideline for treatment of laryngeal cancer, provide evidence-based recommendations for the treatment of laryngeal cancer [21].

In this article, we demonstrate the development of a module for Synthea with the clinical use case of laryngeal cancer. To enable machine learning for decision support, we aim to provide sufficient large sets of training data by introducing a Synthea module for laryngeal cancer. Data will be generated in a real prevalence, in an enriched prevalence, and in a cohort setting, with latter to enable realistic overall, gender-, age-, and stage-specific survival analysis.

## 2. Materials and Methods

### 2.1. Generating Patient Datasets Using Synthea Engine

The architecture of Synthea is shown in Figure 1. According to Walonoski et al. [9], in Synthea the development of a disease, its course, and medical conditions are simulated in a specific module, written in a Generic Module Framework. The framework encodes the module as a state transition machine in a JSON format. The modules are based on guidelines, articles, and databases. A synthetic world is initialized using customizable demographics (e.g., age, gender, living place of the simulated population) and other configurations. Each simulated person then interacts with the disease module (and any other included modules), which computes state transitions at each time step (one week in our configuration) in the synthetic world. At each time step, each module decides whether an event occurs (e.g., throat cancer yes/no decided by our module). Finally, the exporter creates a patient record for each person [9].

The Synthea community already provides several disease modules, which are available on Github (online version control system for software development projects). New modules can be programmed using Synthea’s Generic Module Builder, which we used to create a new module for laryngeal cancer. It is based on a graphical representation of clinical pathways. All mentioned clinical concepts, i.e., risk factors, symptoms, diagnoses, therapies, are represented in states within the graphical network. Edges connect those concepts that contain possible transitions. These transitions also include different distributions, dependencies, or probabilities of occurrence. This allows real-world clinical causalities to be implemented in the modules. An example of the graphical representation of a part of the module and the description of the included transitions can be found in the Appendix A.

When synthesizing a dataset, the Synthea engine runs each generated person from the target population through each installed module. Based on defined disease probabilities, decisions are made about whether a person will develop a disease, and if so, which disease, therapies, and survival outcomes will be modulated. If a patient dies before the current date, he is sorted out, but kept as a record. The engine stops after a predefined target population of living persons has been generated. Depending on the number of installed modules, the synthesized living person will show a realistic distribution of all diseases, symptoms, therapies, etc. as defined in the modules. In contrast to other simulation programs, Synthea does not generate strict cohort data, but a representative cross section of (living) patients (“prevalence approach”). Each patient is synthesized individually and independently from other patients, but the total group of generated patients represents a representative sample of currently living patients. To use Synthea in a cohort setting (“incidence approach”), patient records must first be transformed.

### 2.2. Development of the Laryngeal Cancer Synthea Module

In brief, the model was fed with age- and gender-specific disease rates to determine if and at what age a patient will develop laryngeal cancer. In addition, risk factors, symptoms, tumor stage, therapy, survival up to 5 years, and other factors were modeled.

Figure 2 shows the simplified architecture of our module. When a person enters the module, it decides whether the patient will encounter laryngeal cancer. If the patient develops the disease, the characteristics of the tumor are set so that the results of any future diagnosis, therapy, and survival can be computed accordingly. The subsequent course of the disease in the module can be characterized by four key encounters with the healthcare system: 1. First doctor visit: During this visit, a medical history is taken and a physical examination is carried out. With the results of the diagnostic procedures laryngeal cancer is suspected. 2. Second doctor visit: During this visit, further diagnostic procedures are performed to confirm the diagnosis of laryngeal cancer. 3. Initial treatment: Depending on the characteristics of the patient’s tumor, an appropriate therapy is performed. 4. Adjuvant therapy: If necessary, the patient will receive adjuvant therapy such as radiotherapy or chemoradiation. The last step is the survival submodule, which determines if and when the patient will die within the next 5 years.

For each factor used in the module, empirical evidence for frequencies and associations (reference data) was systematically identified by the German S3-guideline for laryngeal cancer; publications were retrieved from literature databases (as PubMed, Google Scholar) and empirical data from cancer registries, mainly from the Cancer Registry Schleswig-Holstein (Table 1). The implemented diagnostic and therapeutic process is mainly based on the German S3 guideline [21]. Unfortunately, not all statements in the guideline or in the relevant articles were supported by empirical data. In such cases, we had to rely on clinical expertise and experience. For example, if the guideline only states that a condition (or treatment) occurs in “isolated cases”, the frequency of the condition was estimated to be 5%. If the guideline described the condition as a “second-line” or “rare” treatment, its frequency was estimated to be 10%. All underlying assumptions are documented within the model.

Depending on data availability, the natural correlation of selected combinations of variables was simulated. The following associations were considered: T stage of TNM by sex, N stage by T stage and gender, M stage by T and N stage, UICC stage and initial treatment by TNM stage, ECOG score by UICC stage, and survival by UICC stage and age group (10 years).

Since we wanted a realistic, transparent, and understandable disease module, we added sources to each node or decision that is modeled. The Synthea module builder provides a free text attribute for each element. We used this to enter the appropriate sources for the modeled entity, i.e., in the form of a DOI, citation or URL. Although this does not affect the data generation itself, our goal was to provide evidence and insight into the modeled entities for future users. The module is published on our Git page [30], where all calculations and transitions are documented in detail.

### 2.3. Generation of Datasets

Two configurations of the laryngeal cancer module have been built. In the first configuration, as requested by the Synthea community, population-based age-specific disease probabilities depending on gender are used, resulting in a realistic prevalence of laryngeal cancer in the synthesized patient group (prevalence dataset C32-1, Table 2). But, as a quite rare disease (prevalence < 10/100,000), only few cases will be generated even in large samples (based on the intrinsic mechanisms of the engine). Therefore, in the second configuration, the age-specific disease probability was artificially increased to generate a dataset with a high frequency but an approximately realistic age distribution of laryngeal cancer patients (enriched prevalence—dataset C32-2). Here, only patients between the ages 40 and 100 have been synthesized and the correct gender distribution was omitted to generate the largest possible number of patients with laryngeal cancer. Dataset C32-2 serves primarily as a repository for laryngeal cancer. To generate realistic laryngeal cancer survival, the synthetic prevalent data are not suitable. Prevalent patients are all alive and deceased patients are sorted by Synthea into a separate bucket. A cohort approach is needed where only patients with laryngeal cancer (alive and dead) are included. We used dataset C32-3 to construct a cohort of laryngeal cancer patients that reflects the true gender and age distribution of laryngeal cancer patients in Germany (based on the cancer registry of the federal state of Schleswig-Holstein) including realistic cancer survival follow-up. From the synthesized living and deceased patients in dataset C32-2, we randomly draw as many laryngeal cancer patients by age and gender as possible, according to the age and gender distribution of laryngeal cancer in Germany, to create dataset C32-3 (cohort approach). Table 2 summarizes the three datasets. All generated datasets are available on our Git page [30].

### 2.4. Validation of the Laryngeal Cancer Module

The cohort dataset (dataset C32-3) was mainly for the validation of the module used. Absolute and relative frequencies were calculated for qualitative variables and mean and standard deviation for quantitative variables. Where appropriate, analyses were stratified by sex, age group, and stage. Subsequently, Kaplan–Meier survival analyses were performed for the entire group and stratified by gender, age, and tumor stage. As an internal control, we compared these results with the dataset from the Cancer Registry Schleswig-Holstein, which was used to define the general tumor characteristics and survival for the module. The dataset consists of anonymized individual tumor data including 3652 laryngeal cancer cases diagnosed between 1997 and 2021. We used the Kolmogorov–Smirnov test for nonparametric data to test whether the distributions in the simulated dataset differed from those in the Schleswig-Holstein Cancer Registry dataset. Survival was tested using the log-rank test. The significance level was set at 1% to account for multiple testing. Statistical analyses were carried out with SPSS 22.

In a second step, the results based on the Synthea-generated data were qualitatively compared with empirically published, mainly aggregated, data from German cancer registries (e.g., German National Cancer Database at the Robert Koch Institute, Cancer Registry of the State of Schleswig-Holstein, Tumor Registry Munich, and others).

## 3. Results

### 3.1. Prevalence Approach: Dataset C32-1

The first synthesized data set (C32-1), which represents a common patient population with a realistic prevalence of laryngeal cancer, includes a total of 2,061,998 patients (2 million living, 61,998 deceased), 1394 (0.067%) who had a diagnosis of laryngeal cancer (alive: 879, deceased: 515) (Table 2). The 25-year prevalence for laryngeal cancer in this dataset is 43.9 per 100,000 persons (women: 11.5 cases per 100,000 (115 cases), men: 76.4 cases per 100,000 (764 cases)). This result shows a good fit for the 25-year prevalence, acquired through a database request from the ZfKD [16], that ranges from 69–77 cases per 100,000 males and 12–13 cases per 100,000 females from 2014 to 2019.

We found that Synthea underestimates the annual incidence of laryngeal cancer in this dataset. A closer look shows that the 1-year incidence approaches the actual incidence with each year closer to the end date of the simulation we get. About five years before the end of the simulation, the number of new cases per year increases to a number slightly below the population-based incidence. In Table 3, the comparison of the average yearly incidence of the ZfKD [31] data from 1999 to 2019 with the average yearly incidence of the five last simulated years from the C32-1 data shows that Synthea is still underestimating the incidences (total: −22.3%, women: −30.8%, men: −21.4%).

### 3.2. Enriched Approach: Dataset C32-2

The second dataset (dataset C32-2) was designed primarily as a case repository to include as many patients with laryngeal cancer as possible, resulting in a population of 366,429 synthesized patients (of which 116,430 patients are deceased). Out of this population 227,643 patients (62.1%) had a laryngeal cancer diagnosis. As noted in the properties for this dataset, the frequency of women and men is almost the same here (49.7% vs. 50.3%). Other variables show quite similar gender-specific relative frequencies to those in dataset C32-1. Older age groups are slightly underrepresented, while younger age groups are overrepresented.

### 3.3. Cohort Approach: Dataset C32-3

The third data set (data set C32-3) was designed to generate a cohort representative of laryngeal cancer in the German population (in terms of gender and age at diagnosis). Using stratified (by age and gender) random sampling of cases from dataset C32-2, 7354 women (14.7%) and 42,653 men (85.3%) were selected (total of 50,007 cases). The distributions of the synthesized variables were compared with data from the Schleswig-Holstein Cancer Registry (reference data for the module) (Table 4). No statistically significant difference was observed between the simulated and cancer registry data.

Qualitative comparison of the synthesized data with external data showed an overall good agreement (Appendix A). Sex and age distribution showed almost perfect agreement with the national German cancer data (RKI, ZfKD) [31]. Tumor staging (T, N, M category) showed a high concordance with published data from the German cancer registries [32,33,34,35] and selected literature [12,36]. The agreement with UICC staging and grading was moderate. Only location and frequency of smokers showed a larger deviation.

Analysis of survival showed plausible results, with better prognosis in lower stage groups, worse prognosis in older age groups, and no difference for gender. Comparison of 1- and 5-year survival with the Schleswig-Holstein Cancer Registry showed good agreement and no statistically significant differences, except for tumor stage T2 (Figure 3, Appendix A). A comparison of survival with external data from several cancer registries also showed high concordance (Appendix A).

## 4. Discussion

Using the open-source software Synthea (V3.3.1, 30 September 2023), we were able to generate a large amount of realistic patient data for laryngeal cancer, based on external evidence from published data sources. The data for each patient with laryngeal cancer includes the entire patient history starting with symptoms, diagnosis, therapy, and follow up. Therefore, the use of Synthea was a viable choice to generate large synthesized datasets. Despite the advantages, some adjustments to the way Synthea generates data were necessary.

Since we want to use the synthesized data in the German context, we decided to focus on the German population and clinical scenarios, and current treatment regulations from Germany. Since Germany has very strict data protection laws in comparison to other countries and long procedures to obtain large amounts of patient data, the use of synthesized data could solve the problem of access to large datasets including individual patient data. Depending on accessibility, we selected the most appropriate data sources, preferably population-based for Germany. However, not all information needed was freely available or accessible. In particular, data on specific interactions or dependencies between two or more variables, such as the frequency of tumor stage by performance status, are not commonly published. Therefore, we had to use data from other geographical areas, which may lead to different distributions than in Germany. Nevertheless, through a comparison with results from German cancer registries [37] or publications from Germany, we were able to show that we can successfully simulate a realistic German population of laryngeal cancer patients. Although the module’s objective is to simulate a German laryngeal cancer population, the synthesized population should be a close approximation to other European and North American countries, particularly in terms of incidence, gender distribution, and survival [38]. As many countries have their own treatment guidelines, the initial treatment in other countries can differ from the course of treatment simulated in the module. However, due to the open-source nature of Synthea, the module could be easily adapted to other treatment guidelines.

Our new module successfully synthesized laryngeal cancer patients with specific symptoms, different age at diagnosis, tumor localization, tumor stage, therapies, and survival. During development, we ensured that all newly introduced variables had a coherent distribution with respect to the data source (Table 1). However, to prove that we can create datasets which are representative of Germany, we choose to validate our datasets with publications and empirical data from (mainly) Germany that were not used to build the module. Thus, the reasons for the discrepancies in tumor location and smoking prevalence could be that our source publications (see Table 1) use outdated (but most recent) or unrepresentative data. In addition, the validation data for smoking were collected in Greece. It is likely that the smoking habits of laryngeal cancer patients and the prevalence of smokers and ex-smokers may differ from those in Germany, causing the observed discrepancies.

When analyzing the incidence of laryngeal cancer in the original synthetic dataset by year of diagnosis, we found that Synthea underestimates the number of new cases per 100,000 persons the further back in time you go. Nevertheless, the prevalence of laryngeal cancer in the initial synthetic dataset is almost as expected (KID), but the prevalence in older age groups is slightly lower than expected. These underestimations are caused by Synthea’s retrospective functionality, where the starting point is the generation of the patient’s history from presence for a current population with a specific demographics. Let us say you need to generate 100,000 people alive in 2022, and in the target population 0.1% of the population is exactly 100 years old, Synthea will generate a person born in 1922 who will go through all the modules. For example, if such a person is diagnosed with laryngeal cancer at the age of 63, it is statistically unlikely that the patient will live to the age of 100 (due to the worse prognosis of the cancer and other causes of death). If a person dies before 2022, a new person born in 1922 will be generated. This will continue until 100 persons alive in 2022 (=100 years old) are generated. It is quite unlikely (but not impossible) that this person will have a history of laryngeal cancer. This leads to the effect that the number of laryngeal cancer cases in older patients is quite low. From a prevalence point of view, this may be realistic, but from an incidence point of view, the numbers are too low, because simulated patients with laryngeal cancer who died before the target year are excluded (but kept in the separate group of deceased persons). As a result, the synthesized group cannot be used to analyze incidence back in time. Incidence is only meaningful within the most recent simulated years, because in this case only a few incident cases died before the target year. However, these incident cases are not useful for survival analysis because they are all still alive. To overcome this issue, we built a cohort of laryngeal cancer patients outside of Synthea, but using Synthea generated patients. Using a realistic age and sex distribution at the time point of diagnosis random laryngeal cancer cases from the initial synthetized persons (including deceased patients) were drawn.

Since survival is also highly dependent on the patient’s age at diagnosis, it is important for the dataset to have a realistic age distribution while performing survival analyses. To overcome the problems named above we decided to create an age- and gender weighted random sample from the synthesized patients.

If our synthesized data for laryngeal cancer patients is now used for evaluation or AI training, e.g., for survival, such analyses will also lead to realistic results. This is a decisive advantage over other technologies for generating synthetic data, which often cannot represent complex dependencies in the synthetic data body. Particularly for the development of AI algorithms, it is important that the data are fundamentally realistic. In principle, Synthea could be used to refine the data body almost arbitrarily and to model further dependencies. However, this becomes increasingly complex as the level of detail increases. For example, the current module for simulating survival considers age in four groups and tumor stage in five characteristics. This means that for a realistic 5-year survival (with one data point per year), 100 node probabilities must be determined and entered into the model. If you also want to simulate survival depending on the type of therapy, hundreds or thousands of additional node points must be added. Even if this is technically feasible, the corresponding probabilities for the nodes must first be determined from large and high-quality data sources (e.g., cancer registries with clinical data).

Building the module manually required significant resources, including guideline and literature research by clinical experts, manual creation of the module using the Module Builder, and test case generation and validation. However, this expert-based approach ensures that all modeled entities and decisions are medically sound and that clinical evidence is available. Of course, the module is ready for use by clinical researchers or medical data scientists. Maintenance of the laryngeal cancer module, on the other hand, would require a similar amount of effort. Individual nodes or edges in the graph representation would need to be updated or added, syntactic correctness would need to be ensured, and the updated module would also need to be validated. In fact, the community aspect of the Synthea system would allow users to customize their needs (e.g., regional differences in policies), while the base module would be made available to everyone through the official Synthea channels.

We aim to incorporate the laryngeal cancer module into the Synthea lifecycle modules. This will involve matching the demographic characteristics to their internal Massachusetts Census data. We believe that our contribution helps to provide more insight into realistic data synthetization for laryngeal cancer. In the next steps, we plan to add more dependencies to survival to make it even more attractive for AI training. New versions will be made available on our Git page.

## 5. Conclusions

In this article, we presented a feasible approach to create a laryngeal cancer module using Synthea, the Synthetic Patient Population Simulator. We were able to demonstrate the necessary steps to generate realistic electronic patient records for the clinical use case of laryngeal cancer, a quite rare oncological disease. Furthermore, with Synthea we have been able to produce a high number of patients with laryngeal cancer with a realistic distribution of gender, age, risk factors, and survival. By transforming the synthesized data, we were able to create a dataset containing a large number of only patients with laryngeal cancer that was representative of age, sex, survival, and other factors. Currently, the synthesized data form module is used for further analyses and research at the University of Leipzig. Work is ongoing using the module to train an AI that can predict death from laryngeal cancer. Our module will be further optimized for this purpose. In the development of clinical decision support for head and neck cancer, the synthetic data should provide good opportunities for the evolution of AI applications, particularly since the results of such research will be close to real-world data, even though only synthetic data are used. Using Synthea, real-world data can be partially outperformed because the limitation of small case numbers (as in laryngeal cancer) does not apply to synthesized data. This is an important precondition in the development of AI applications.

## Figures and Tables

**Figure 1 cancers-16-00639-f001:**
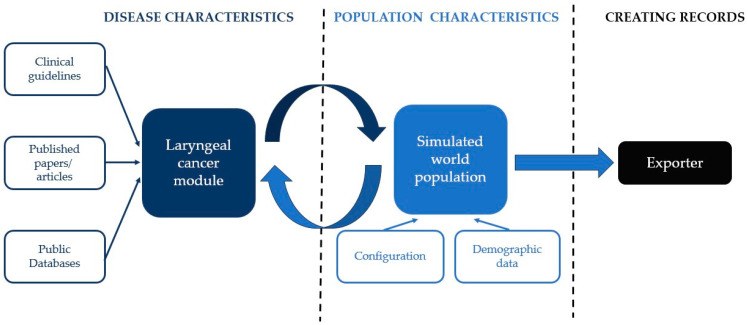
Synthea software (V3.3.1, 30 September 2023) architecture (adapted according to Walonoski et al. [9]).

**Figure 2 cancers-16-00639-f002:**
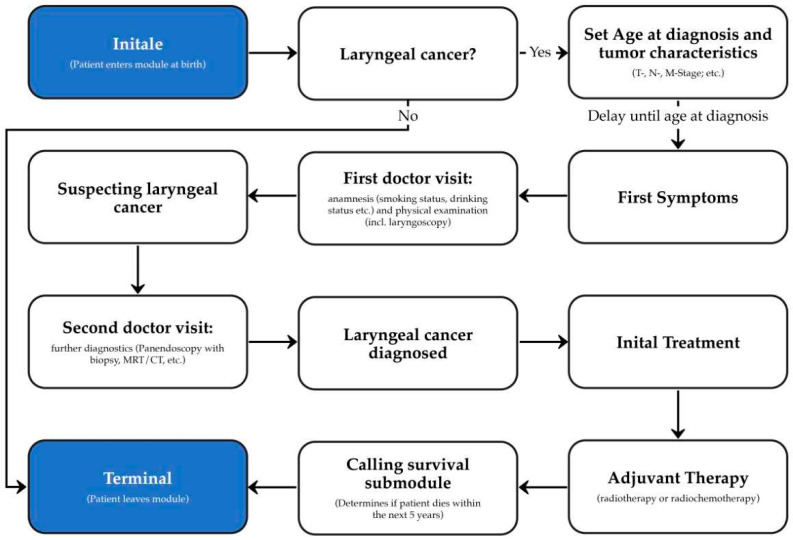
Schematic architecture of the laryngeal cancer module for Synthea.

**Figure 3 cancers-16-00639-f003:**
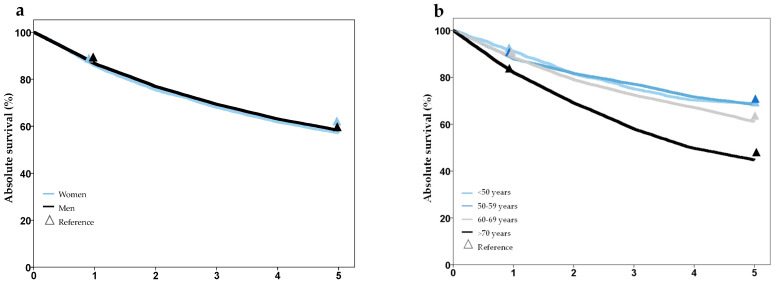
Survival after laryngeal cancer stratified by gender (**a**), age group (**b**), T-stage (**c**), and UICC-stage (**d**), based on synthesized cohort data (dataset C32-3). Triangles show reference values from the Cancer Registry Schleswig-Holstein.

**Table 1 cancers-16-00639-t001:** Reference data: Summary of the mainly used variables and associated data sources.

Variable	Description	Source
Age, gender	Age distribution of laryngeal cancer patients in 5-year groups reflecting gender- and age-specific probability of disease, age <40 years was excluded	Cancer Registry SH
Performance status (ECOG)	Frequency of ECOG score depending on UICC (0–5)	Manigreeva Krishnatreya, 2014 [22]
Alcohol	Frequency of alcohol consumption in laryngeal cancer patients (no, occasionally, regular, excessive and former excessive alcohol consumption)	Engel, 2018 [23]
Smoking	Prevalence of smokers, non-smokers and former smokers in laryngeal cancer patients (non-smoker, former smoker, active smoker)	Engel, 2018 [23]
Human papilloma virus (HPV)	Frequency of HPV-associated laryngeal cancer (Yes/No)	Castellsagué, 2016 [24]
Glottic Precursor Lesion	Occurrence and differentiation of laryngeal precursor lesions prior to development of laryngeal cancer (carcinoma in situ, NOS, mild, moderat, severe)	Sanninoa, 2020 [25]
Symptoms	Frequency of symptoms (hoarseness, otalgia, dysphagia, insomnia, mouth symptoms, chest infection, sore throat and dysphonia) that onset up to one year before diagnosis of laryngeal cancer (Yes/No)	Shephard, 2019 [26]
T staging	Distribution of tumor size (T category of TNM) depending on gender (T1–T4, including T1a, T1b, T4a and T4b)	Cancer Registry SH
N staging	Distribution of regional lymph node involvement (N category of TNM) depending on gender and T staging (N0–N3)	Cancer Registry SH
M staging	Distribution of distant metastasis (M category, TNM) depending on gender, T and N stage (M0, M1)	Cancer Registry SH
UICC	Condensed tumor staging, compiled form T, N, M according UICC (I–IV)	Brierly, 2016 [27]
Grading	Frequency of levels of histopathological differentiation in laryngeal cancer cells (G1–G3)	Ketterer, 2020 [28]
Localisation of primary tumor	Distribution of primary cancer sites (glottis, subglottis, supraglottis)	German S3 guideline for laryngeal cancer [21]
Localisation of metastasis	distribution of the localisation of laryngeal cancer metastasis (lung, bone, skin, central nervous system, other organs)	Spector, 2001 [29]
Initial treatment	Frequency of initial treatments depending on TNM and UICC staging (open partial laryngectomy, transoral partial laryngectomy, laryngectomy, radiotherapy, chemotherapy, radio-chemotherapy)	Engel, 2018 [23], German S3 guideline for laryngeal cancer [21]
R-status	Frequency of resection status in surgically treated (R0–R2)	Engel, 2018 [23]
Adjuvant therapy	Frequency for a postoperative treatment (postoperative radiotherapy, postoperative radio-chemotherapy, no postoperative therapy)	German S3 guideline for laryngeal cancer [21]
Neck dissection	Operation including regional lymph node (Yes (curative or elective)/No)	German S3 guideline for laryngeal cancer [21]
Survival	1-, 2-, 3-, 4-, 5- year overall survival (%) stratified by age, gender and T, N, M stage	Cancer Registry SH

ZfKD: German Center for Cancer Registry Data at the Robert Koch-Institute, Cancer Registry SH: Cancer Registry of the German federal state Schleswig-Holstein, Dataset laryngeal cancer 1997–2021, August 2022, TNM: International Tumor Classification [27].

**Table 2 cancers-16-00639-t002:** Overview of synthesized data set with different approaches.

	Dataset C32-1	Dataset C32-2	Dataset C32-3
Description	Population-based prevalence	Enriched prevalence	Cohort approach (age and gender matched)
Patients synthetized	alive: 2,000,000 (total: 2,061,998)	alive: 249,999 (total: 366,429)	alive: 27,095 (total: 50,007)
… with laryngeal cancer (ICD10: C32)	alive: 879 (total: 1394)	alive: 98,084 (total: 227,643)	alive: 27,095 (total: 50,007)
Gender (in patients with laryngeal cancer) women/men	173/1221	113,316/114,327	7354/42,653
Age at diagnosis (Mean, SD)	60.9 (±10.5)	63.1 (±10.2)	65.5 Years (±10.6)

**Table 3 cancers-16-00639-t003:** Incident cases per 100,000 in the Synthea data (C32-1) in comparison to German national data [31] (retrieved April 2023).

	Total	Women	Men
Synthea data (Average of the last five simulated years)	3.63	0.83	6.51
ZfKD data (average from 1999–2019)	4.68	1.2	8.3
Deviation%	−22.3%	−30.8%	−21.4%

**Table 4 cancers-16-00639-t004:** Comparison of synthesized data (Synthea) with data from cancer registry Schleswig-Holstein (SH, 2000–2021), “n. s.” indicates that no significant difference between both datasets is present.

		Synthea		CR SH		
Variable		Female	Male	Female	Male	*p*-Value
Gender (%, row)		14.7	85.3	15.8	84.2	n. s.
Age (%)	40–49	8.7	6.1	9.4	6.6	n. s.
	50–59	24.0	23.7	23.6	23.1	
	60–69	32.9	35.1	32.8	35.0	
	70–79	22.1	25.0	21.9	25.2	
	80+	12.3	10.1	12.3	10.1	
T-Staging (%)	T1	36.4	43.2	43.2	43.3	n. s.
	T2	23.5	21.0	21.0	21.0	
	T3	23.9	18.9	18.9	18.5	
	T4	16.2	16.9	16.9	17.1	
N-Staging (%)	N0	67.3	73.9	66.6	72.0	n. s.
	N1	7.9	7.0	9.2	7.6	
	N2	22.0	16.5	22.4	17.7	
	N3	2.8	2.6	2.9	2.7	
M-Staging (%)	M0	94.6	95.6	93.9	95.6	n. s.
	M1	5.4	4.4	6.1	4.4	
UICC Staging (%)	I	34.9	40.5	35.1	41.4	n. s.
	II	15.4	14.5	15.5	14.7	
	II	15.1	15.1	15.3	15.3	
	IV	34.6	29.9	34.1	34.1	
Grading (%)	1	7.3	6.7	8.3	8.7	n. s.
	2	68.9	69.6	62.5	67.8	
	3	23.0	23.7	29.2	23.5	
Survival (%)	1-year	85.3	86.5	85.7	86.9	n. s.
	5-year	57	58.1	63.6	58.9	
Survival 5-year (%) by UICC	I	76.3	75.3	80.2	76.9	n. s.
	II	67.4	64.0	76.2	67.9	
	II	58.0	56.0	67.4	58.9	
	IV	35.5	33.5	44.8	35.8	

## Data Availability

Data are contained within the article and Appendix A are openly available in our institutional repository (https://git.iccas.de/synthea/laryngeal-cancer (accessed on 29 January 2024)).

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
