# Peer review of "Generation of a Realistic Synthetic Laryngeal Cancer Cohort for AI Applications"

_cancers, 2024, doi:10.3390/cancers16030639_

Round 1

Reviewer 1 Report

Comments and Suggestions for Authors

The manuscript is well-written, and the topic is of interest. However, several improvements can be applied to enhance the quality of the manuscript.

1. The introduction lacks a brief description of related work specifically on synthetic patient generation. I recommend incorporating a discussion and comparison of synthetic data generation models used in similar applications. Consider referencing a couple of reviews on the generation of synthetic data in medical applications:

https://doi.org/10.1016/j.neucom.2022.04.053

https://doi.org/10.1016/j.cosrev.2023.100546

- https://paperswithcode.com/paper/methods-for-generating-and-evaluating

2. Provide a more robust justification for choosing Synthea over other synthetic data generation models. While some reasons are mentioned in the conclusion, it would be beneficial to state this in the introduction as well.

3. Revise the numbering of subsections in Section 2 for clarity.

4. Consider adding a complementary figure or pseudocode in Section 2.1 to improve understanding of the patient dataset generation process using the Synthea engine.

5. In Section 2.2, provide an explanation for the chosen distributions. Specifically, clarify the reasoning behind estimating 5% and 10% frequencies based on treatment choice.

6. Address the absence of results from Dataset C32-2 in Section 3.2. Include a table with the results or provide a clear explanation for excluding this table.

7. Ensure that tables are referenced in the text for better coherence.

8. In Table 4, the categorization in the Fit column appears subjective. Clarify the methodology used to determine Good or Bad fits.

9. If feasible, consider performing survival analyses with real data and the other datasets to provide additional evidence for evaluating your module.

In conclusion, I recommend accepting the manuscript after implementing these improvements.

Comments on the Quality of English Language

There are some typos on the spelling:

- Lines 289 and 347: Especially -> Specially.

Some phrases are quite extensive to understand. I suggest to make smaller sentences.

Author Response

Reviewer’s Comments:
Reviewer 1:

Comment of reviewer:

            English language and style: Minor editing of English language required

There are some typos on the spelling:

- Lines 289 and 347: Especially -> Specially.

Some phrases are quite extensive to understand. I suggest to make smaller sentences.

Response to reviewer:

We conducted a check and correction of spelling and style wherever appropriate. We did not mark all changes in the correction mode, because it would make the manuscript almost unreadable. Significant changes and answers to the reviewer were marked in yellow.

Comment of reviewer:

The manuscript is well-written, and the topic is of interest. However, several improvements can be applied to enhance the quality of the manuscript.

In conclusion, I recommend accepting the manuscript after implementing these improvements.

Response to reviewer:

Thank you for your kind assessment. We made several changes in the manuscript.

Comment of reviewer:

  1. The introduction lacks a brief description of related work specifically on synthetic patient generation. I recommend incorporating a discussion and comparison of synthetic data generation models used in similar applications. Consider referencing a couple of reviews on the generation of synthetic data in medical applications:

- https://doi.org/10.1016/j.neucom.2022.04.053

- https://doi.org/10.1016/j.cosrev.2023.100546

- https://paperswithcode.com/paper/methods-for-generating-and-evaluating

Response to reviewer:

Thank you for pointing out the lacks of related work. We added a paragraph to the introduction section, relating our work to other existing technologies and added the references to the introduction of the manuscript.

Comment of reviewer:

  1. Provide a more robust justification for choosing Synthea over other synthetic data generation models. While some reasons are mentioned in the conclusion, it would be beneficial to state this in the introduction as well.

Response to reviewer:

Thank you for this valuable remark. We added clarification to the introduction section.

Comment of reviewer:

  1. Revise the numbering of subsections in Section 2 for clarity.

Response to reviewer:

            Thank you for this valuable remark, the numbering is corrected.

Comment of reviewer:

  1. Consider adding a complementary figure or pseudocode in Section 2.1 to improve understanding of the patient dataset generation process using the Synthea engine.

Response to reviewer:

Thank you for that point. We created a figure showing the architecture of Synthea and added some explaining text.

Additionally, according to reviewer 2, we have created an additional figure that illustrates the rough sequence of the simulation of a patient with laryngeal carcinoma. We have explained the simulation process further in the text. The original figure with visualized Synthea code was moved to the supplement, and a detailed description was added.

Comment of reviewer:

  1. In Section 2.2, provide an explanation for the chosen distributions. Specifically, clarify the reasoning behind estimating 5% and 10% frequencies based on treatment choice.

Response to reviewer:

This is an important point. Unfortunately, even in S3-guidelines or in published papers not all information needed for a Synthea module (as treatment frequencies) are available. Often guidelines use terms as “individual choice for treatment” or “treatment can be applied” without stating any frequencies. For such missing information, assumptions must be made. In our module such assumptions are based on the clinical experience from the clinical coauthors, which are experts in the field of laryngeal cancer. Whenever an assumption was made, it was documented in the module.

Comment of reviewer:

  1. Address the absence of results from Dataset C32-2 in Section 3.2. Include a table with the results or provide a clear explanation for excluding this table.

Response to reviewer:

The C32-2 dataset is primarily used as a repository to enrich laryngeal carcinoma cases in order to have sufficient case numbers for the cohort dataset later on. This is done by increasing the risk of getting laryngeal cancer so that more people from the population fall ill. The distributions for the simulated variables are not significantly changed by this procedure. We added this information now to the article, to clarify that C32-2 is more an intermediate step to the cohort model. But still, dataset C32-2 could be used when a high number of unselected laryngeal cancer patients is needed.

We added to section 2.3.: “Dataset C32-2 serves in the first line as a repository for laryngeal cancer.”
and to section 3.2: “The second dataset (dataset C32-2) was mainly designed as a case repository to let as many patients as possible suffer from laryngeal cancer” ...

and “Other variables show quite similar gender-specific relative frequencies as in dataset C32-1.”

Comment of reviewer:

  1. Ensure that tables are referenced in the text for better coherence.

Response to reviewer:

Thank you for pointing this out. We carefully checked that all tables are reference in a proper manner.

Comment of reviewer:

  1. In Table 4, the categorization in the Fit column appears subjective. Clarify the methodology used to determine Good or Bad fits.

Response to reviewer:

Initially, we have chosen a deterministic approach to estimate how well the synthetic data set match real-world data from different sources. An absolute deviation of 5% points was assumed to be a good match. This corresponds to (usually random) deviations that can be observed between different cancer data sets. Absolute deviations from 5% up to 10% were considered as moderate. Larger deviations are labelled as poor fit.

Reviewer 2 also mentioned this point.

We have therefore fundamentally reconsidered this point. We focus now more on the point whether the simulated cohort dataset shows comparable distributions to the essential data used for building the module (mainly taken from Cancer Registry Schleswig-Holstein, CRSH). It is important to show that Synthea will generate realistic data. For this, we compared the distributions of selected variables from the simulated data set with the CRSH data for statistically significant differences. We used Kolmogorov-Smirnow-Test (a non-parametrical test for nominal and ordinal variables) and log-rank test for survival with the null hypothesis, that there is no difference between the distributions. Significance value was set to 1%, to consider multiple testing.

The comparisons showed that there are no relevant significant differences between the original data and the simulated data. 

In the article, we moved the already existing supplementary table (old S1), comparing results from the module with CRSH, to the main article and added results of significance testing. The initial table with the comparison to other data sources was moved to the supplement, the initial assessment (good, moderate, bad fit) was removed, and differences are now only reported in a qualitative manner in the text.

The initial table with the comparison to other data sources was moved to the supplement, the initial assessment (good, moderate, bad fit) was removed, and differences are now only reported in a qualitative manner in the text.

Comment of reviewer:

  1. If feasible, consider performing survival analyses with real data and the other datasets to provide additional evidence for evaluating your module.

Response to reviewer:

Thank you for mentioning that. We now calculated survival with the CRSH data (used to set up the module). The fit between simulated and real survival is almost perfect.

We added a column with the CRSH data to supplementary table 2. In the figure 3, we replaced the references with the CRSH survival result, so now showing survival for the simulated and the used, real data.

Reviewer 2 Report

Comments and Suggestions for Authors

This manuscript exposes an interesting and innovative approach to facilitate the application of AI algorithms for laryngeal cancer analysis. This is performed by using synthetic data, so computer-aided decision systems can be developed without concerning patient’s privacy. The authors have validated the obtaining results against real data, demonstrating that generated synthetic data generally replicates the real data in almost everything generated variables. However, before publication, I would suggest some changes:

11)      Some references regarding examples of the positive impact that the use of synthetic data has in AI models performance (preferable using EHR, or tabular data, in general), would give strength to the justification of the work.

22)      When Synthea is introduced, it is stated that “the tool uses a combination of algorithms”. Which algorithms? How does this tool work? Does It use a statistical approach, Machine Learning techniques? Either in the “Introduction” or in the “Material and Methods” section, a deeper and detailed technical explanation would provide the reader valuable information to fully understand the tool you used.  

33)      From line 87 to 92, some works that used Synthea were cited. However, for the sake of completeness, it would be worth noting which was the impact (if any) of using synthetic data in the obtained results of those works.

44)      At the end of the introduction, the contribution of this paper is stated. However, it is mixed with the exposition of some epidemiological data. As this is part of the use of case wanted to be addressed, this should be placed earlier in the introduction, leaving the last paragraph to introduce your work itself. A clearer exposition of your contribution in these last paragraphs would benefit the reader.

55)      Related to 2), a more technical explanation of what has been done with Synthea and the designed module is required. It is the main part of the work, so more details are needed so the reader can fully understand your valuable contribution. For example, Figure 1 is cited but not explained, yet it is one of the main parts of your work.

66)      In line 155, “standard Synthea settings and demographics” are mentioned. However, the reader that is not familiar with this tool does not know what does it mean. An explanation of this standard configuration would avoid doubts in this matter.

77)      Even the code repository is mentioned and cited, and it is truly positive as a contribution of your work, the relevant calculations and transitions should be properly stated and explained in detail within the text. The information contained in the repository should be a complementary material rather than the place to find the explanation of the basis of this work.

88)      In line 209, it is said that “We did not perform any statistical tests, because due to the large sample size even small differences would become significant.” If it is significant, how this would impact in AI? To train DL models, if synthetic data does not represent reality, results cannot be applied to reality. Please, give a proper justification so this issue can be clarified.

99)      Regarding the margins of the observed deviations, is there any work that used them? Or is by clinical criteria? Please, clarify it. Besides, which statistical analyses were done? They are only mentioned, but not explained.

110)   Regarding the results, there are a few things that remain unclear. Why is it only the cohort synthetic dataset (C32-3) validated and not all of them? It is not clear the contribution of the other two datasets. A proper justification should be given. Besides, Was the C32-3 dataset generated with Synthea, or is it just a subset of the C32-2 after “random drawing of cases”? Please, clarify this.

111)  Data from 32-2 is not shown. It is hard to see what this contributes to the work.

112)   What does “Good”, “moderate” and “bad” fit mean? Which analysis was performed? From the text it is not clear. Since this is the criteria followed to conclude that Synthea has worked effectively in this case, I kindly suggest providing more details on this matter. If it is only differences in the standard deviation, I kindly suggest also to perform a deeper analysis.

113)  When several validation sets are used for one variable, where does the data come from? Is it an average? Is It the same value for all references? Please, indicate it when necessary.

114)   Even if it is said that smokers show bad fit (and results show so), it appears in the table that fit is “good”.

115)   Even results are clear in the Figure 2, some quantification about the differences in the reference and the obtained value is missed in the text (even they are place in Table S2). I suggest moving Table S2 into de manuscript, since they are results that directly explain how good has behaved your experiment. Again, more statistics, parameters would enrich the contribution of this manuscript (bar diagramas, histograms of the variables, skewness and kurtosis of the continuous variables, etc.). Besides, are there references of the curves to compare to, and not only the absolute survival values, but the whole curve?

116)   In the discussion “nodes” and “probabilities” are mentioned for the first time in the text. But their relationship with the Synthea model, how are they configured/generated, etc., has not been mentioned. Please, to better understand your work and the implications of such details in the final results, explain these concepts in the “Materials and methods” section.  

117)   Since this is a special issue on AI, and AI is explicitly mentioned in the title, it should be given more importance in the text. Issues such as if the C32-3 dataset is suitable for an AI training should be analysed. Would be the AI model a regression model or a classification model? Have been done some preliminary tests with simple Machine Learning models to test its suitability? If these experiments have not been done, it would be advisable to add some future lines, or to give some indications of how data scientists or AI-researchers could work with your provided dataset.

Besides, some minor comments:

a1)       Page 1, line 50 (and following). There are capital letters after commas, where lower case should be placed.

b2)      Resolution of Figure 1 should be increased.

c3)       Within the text there are some expressions such as “represents a representative”, “population of persons”, “it’s”, etc. Please, seeking clarity and concise language replace them.

d4)      Please review the way the tables are cited within the text.

Comments on the Quality of English Language

There are no comments regarding the Quality of English Language.

Author Response

Reviewer 2:

Comment of reviewer:

            English language and style: Minor editing of English language required

Response to reviewer:

We conducted a check and correction of spelling and style wherever appropriate. We did not mark all changes in the correction mode, because it would make the manuscript almost unreadable. Significant changes and answers to the reviewer were marked in yellow.

Comment of reviewer:

This manuscript exposes an interesting and innovative approach to facilitate the application of AI algorithms for laryngeal cancer analysis. This is performed by using synthetic data, so computer-aided decision systems can be developed without concerning patient’s privacy. The authors have validated the obtaining results against real data, demonstrating that generated synthetic data generally replicates the real data in almost everything generated variables. However, before publication, I would suggest some changes:

Response to reviewer:

We are grateful for your assessment and made several changes in the manuscript.

Comment of reviewer:

1)      Some references regarding examples of the positive impact that the use of synthetic data has in AI model’s performance (preferable using EHR, or tabular data, in general), would give strength to the justification of the work.

Response to reviewer:

We have added some positive examples in the introduction to demonstrate the impact of synthetic data application.

Comment of reviewer:

2)      When Synthea is introduced, it is stated that “the tool uses a combination of algorithms”. Which algorithms? How does this tool work? Does It use a statistical approach, Machine Learning techniques? Either in the “Introduction” or in the “Material and Methods” section, a deeper and detailed technical explanation would provide the reader valuable information to fully understand the tool you used.

Response to reviewer:

Thank you for that point. We now described the functionality of Synthea more in detail.
We created a figure showing the architecture of Synthea and added some explaining text (according to reviewer 1). Secondly, we have created an additional figure that illustrates the rough sequence of the simulation of a patient with laryngeal carcinoma. We have explained the simulation process further in the text. The original figure with visualized Synthea code was moved to the supplement and a detailed description was added.

Comment of reviewer:

3)      From line 87 to 92, some works that used Synthea were cited. However, for the sake of completeness, it would be worth noting which was the impact (if any) of using synthetic data in the obtained results of those works.

Response to reviewer:

      Thank you for the suggestion. We added further information on Synthea impact specifically.

Comment of reviewer:

4)      At the end of the introduction, the contribution of this paper is stated. However, it is mixed with the exposition of some epidemiological data. As this is part of the use of case wanted to be addressed, this should be placed earlier in the introduction, leaving the last paragraph to introduce your work itself. A clearer exposition of your contribution in these last paragraphs would benefit the reader.

Response to reviewer:

Thank you for that remark. We restructured the introduction to have a clearer description of the clinical use case and finally a clear exposition of our own contributions.

Comment of reviewer:

5)      Related to 2), a more technical explanation of what has been done with Synthea and the designed module is required. It is the main part of the work, so more details are needed so the reader can fully understand your valuable contribution. For example, Figure 1 is cited but not explained, yet it is one of the main parts of your work.

Response to reviewer:

We agree that the functionality of Synthea must described more in detail. As stated in point 2 we now added a common description of Synthea. The architecture of our new module was worked out more in detail.

Comment of reviewer:

6)      In line 155, “standard Synthea settings and demographics” are mentioned. However, the reader that is not familiar with this tool does not know what does it mean. An explanation of this standard configuration would avoid doubts in this matter.

Response to reviewer:

Thank you. We removed that statement for the standard settings, since it does not contribute to method description.

Comment of reviewer:

7)      Even the code repository is mentioned and cited, and it is truly positive as a contribution of your work, the relevant calculations and transitions should be properly stated and explained in detail within the text. The information contained in the repository should be a complementary material rather than the place to find the explanation of the basis of this work.

Response to reviewer:

Thank you for this remark. We have added some explanation of the calculations and transitions in our Synthea module, now moved to the supplement. We believe; however, this can only be done exemplary because the complexity of our module is higher than can be described in sufficient detail in the manuscript without compromising readability and comprehensibility for the reader. We gladly provide the original code to have it examined by any colleague in the community.

Comment of reviewer:

8)      In line 209, it is said that “We did not perform any statistical tests, because due to the large sample size even small differences would become significant.” If it is significant, how this would impact in AI? To train DL models, if synthetic data does not represent reality, results cannot be applied to reality. Please, give a proper justification so this issue can be clarified.

Response to reviewer:

Thank you for mentioning this point. Reviewer 1 also had suggestion on the validation of the module.

We have therefore fundamentally reconsidered this point. We focus now more on the point whether the simulated cohort dataset shows comparable distributions to the essential data used for building the module (mainly taken from Cancer Registry Schleswig-Holstein, CRSH). It is important to show that Synthea will generate realistic data. For this, we compared the distributions of selected variables from the simulated data set with the CRSH data for statistically significant differences. We used Kolmogorov-Smirnow-Test (a non-parametrical test for nominal and ordinal variables) and log-rank test for survival with the null hypothesis, that there is no difference between the distributions. Significance value was set to 1%, to consider multiple testing.

The comparisons showed that there are no relevant significant differences between the original data and the simulated data. So, one can consider, that the Synthea generated data is comparable to the real data.

In the article, we moved the already existing supplementary table (old S1), comparing results from the module with CRSH, to the main article and added results of significance testing. The initial table with the comparison to other data sources was moved to the supplement, the initial assessment (good, moderate, bad fit) was removed, and differences are now only reported in a qualitative manner in the text.

Comment of reviewer:

9)      Regarding the margins of the observed deviations, is there any work that used them? Or is by clinical criteria? Please, clarify it. Besides, which statistical analyses were done? They are only mentioned, but not explained.

Response to reviewer:

This point has been answered by the revision (see point 8).

Comment of reviewer:

10)   Regarding the results, there are a few things that remain unclear. Why is it only the cohort synthetic dataset (C32-3) validated and not all of them? It is not clear the contribution of the other two datasets. A proper justification should be given. Besides, Was the C32-3 dataset generated with Synthea, or is it just a subset of the C32-2 after “random drawing of cases”? Please, clarify this.

Response to reviewer:

The C32-2 dataset is primarily used as a repository to enrich laryngeal carcinoma cases in order to have sufficient case numbers for the cohort dataset later on. This is done by increasing the risk of getting laryngeal cancer so that more people from the population fall ill. The distributions for the simulated variables are not significantly changed by this procedure. We added this information now to the article, to clarify that C32-2 is more an intermediate step to the cohort model. But still, dataset C32-2 could be used when a high number of unselected laryngeal cancer patients is needed.

We added to section 2.3.: “Dataset C32-2 serves in the first line as a repository for laryngeal cancer.”
and to section 3.2: “The second dataset (dataset C32-2) was mainly designed as a case repository to let as many patients as possible suffer from laryngeal cancer” ...

and “Other variables show quite similar gender-specific relative frequencies as in dataset C32-1.”

The cohort dataset (C32-3) could not be created directly with Synthea, because Synthea generate only prevalent patient populations (all patients alive) and an additional bucket with deceased persons. To generate a cohort (to enable survival analyses), all simulated cases (alive and death) were used to draw the cohort members (age and gender matched to the real distribution of laryngeal cancer patients at diagnosis).
We tried to describe this now better in the article.

Comment of reviewer:

11)  Data from 32-2 is not shown. It is hard to see what this contributes to the work.

Response to reviewer:

Please refer to point 10.

Comment of reviewer:

12)   What does “Good”, “moderate” and “bad” fit mean? Which analysis was performed? From the text it is not clear. Since this is the criteria followed to conclude that Synthea has worked effectively in this case, I kindly suggest providing more details on this matter. If it is only differences in the standard deviation, I kindly suggest also to perform a deeper analysis.

Response to reviewer:

Please refer to point 8. With the revision the assessment was removed and statistical testing was introduced.

Comment of reviewer:

13)  When several validation sets are used for one variable, where does the data come from? Is it an average? Is It the same value for all references? Please, indicate it when necessary.

Response to reviewer:

Thank you for this point. When more than one validation data source was used, we calculated mean and range. This was added to the table (now supplementary table S1)

Comment of reviewer:

14)   Even if it is said that smokers show bad fit (and results show so), it appears in the table that fit is “good”.

Response to reviewer:

Thank you for revealing this error. As the assessment was removed, the error also was removed.

Comment of reviewer:

15)   Even results are clear in the Figure 2, some quantification about the differences in the reference and the obtained value is missed in the text (even they are place in Table S2). I suggest moving Table S2 into de manuscript, since they are results that directly explain how good has behaved your experiment. Again, more statistics, parameters would enrich the contribution of this manuscript (bar diagramas, histograms of the variables, skewness and kurtosis of the continuous variables, etc.). Besides, are there references of the curves to compare to, and not only the absolute survival values, but the whole curve?

Response to reviewer:

Thank you for this important point. We understand the intention and mainly comply with this suggestion. As stated for point 8 we now focus more on the comparison of the simulated data set with the dataset that was mainly used for building the module. We did statistical testing for the comparisons. In the survival figure (old figure 2, now 3) we added one- and five-year survival of the reference data used for building the model (adding all reference would have overloaded the figure). Survival was tested with log-rank test which compares the whole curves. Table S1 was moved to main article to better show the comparison between simulated and reference data. We also reviewed these distributions graphically. But we already placed three figures and four table in the article, and did not want to overload the article with more figures.
We are confident, that we could show now that the simulated data in good agreement with the reference data.

Comment of reviewer:

16)   In the discussion “nodes” and “probabilities” are mentioned for the first time in the text. But their relationship with the Synthea model, how are they configured/generated, etc., has not been mentioned. Please, to better understand your work and the implications of such details in the final results, explain these concepts in the “Materials and methods” section. 

Response to reviewer:

Thank you for pointing that out. We added more details in the methods section, also introducing those terms.

Comment of reviewer:

17)   Since this is a special issue on AI, and AI is explicitly mentioned in the title, it should be given more importance in the text. Issues such as if the C32-3 dataset is suitable for an AI training should be analyzed. Would be the AI model a regression model or a classification model? Have been done some preliminary tests with simple Machine Learning models to test its suitability? If these experiments have not been done, it would be advisable to add some future lines, or to give some indications of how data scientists or AI-researchers could work with your provided dataset.

Response to reviewer:

Thank you for this point. Our module is the foundation for development and training of an AI in laryngeal cancer. The module is currently first time used for this purpose. Unfortunately, no final results of this ongoing work are available. So, no further information on your questions can be given right now.
But we added to the conclusion section: “Currently, the synthesized data form module is being used for further analyses and research at the University of Leipzig. Work is ongoing using the module to train an AI that can predict death from laryngeal cancer. Our module will be further optimized for this purpose in the next steps.”

Comment of reviewer:

Besides, some minor comments:

a1)       Page 1, line 50 (and following). There are capital letters after commas, where lower case should be placed.

b2)      Resolution of Figure 1 should be increased.

c3)       Within the text there are some expressions such as “represents a representative”, “population of persons”, “it’s”, etc. Please, seeking clarity and concise language replace them.

d4)      Please review the way the tables are cited within the text.

Response to reviewer:

Thank you for these comments. We have considered and corrected all points.

Round 2

Reviewer 1 Report

Comments and Suggestions for Authors

The comments have been correctly and adequately fixed. The manuscript is well-written and the quality of it has been improved.

Comments on the Quality of English Language

None
